# Blurred by a “Puff of Smoke”—A Case-Based Review on the Challenging Recognition of Coexisting CNS Demyelinating Disease and Moyamoya Angiopathy

**DOI:** 10.3390/ijms26115030

**Published:** 2025-05-23

**Authors:** Isabella Canavero, Nicola Rifino, Carlo Antozzi, Valentina Caldiera, Elena Colombo, Tatiana Carrozzini, Giuseppe Ganci, Paolo Ferroli, Francesco Acerbi, Benedetta Storti, Giorgio Battista Boncoraglio, Antonella Potenza, Giuliana Pollaci, Gemma Gorla, Elisa Ciceri, Patrizia De Marco, Laura Gatti, Anna Bersano

**Affiliations:** 1Cerebrovascular Unit, Fondazione IRCCS Istituto Neurologico Carlo Besta, 20133 Milan, Italy; tatiana.carrozzini@istituto-besta.it (T.C.); benedetta.storti@istituto-besta.it (B.S.); giorgio.boncoraglio@istituto-besta.it (G.B.B.); antonella.potenza@istituto-besta.it (A.P.); giuliana.pollaci@istituto-besta.it (G.P.); gemma.gorla@istituto-besta.it (G.G.); laura.gatti@istituto-besta.it (L.G.); anna.bersano@istituto-besta.it (A.B.); 2Neuroimmunology and Neuromuscular Diseases Unit, Fondazione IRCCS Istituto Neurologico Carlo Besta, 20133 Milan, Italy; carlo.antozzi@istituto-besta.it; 3Diagnostic Radiology and Interventional Neuroradiology Unit, Fondazione IRCCS Istituto Neurologico Carlo Besta, 20133 Milan, Italy; valentina.caldiera@istituto-besta.it (V.C.); giuseppe.ganci@istituto-besta.it (G.G.); elisa.ciceri@istituto-besta.it (E.C.); 4Multiple Sclerosis Centre, IRCCS Mondino Foundation, 27100 Pavia, Italy; elena.colombo@mondino.it; 5Neurosurgery Department, Fondazione IRCCS Istituto Neurologico Carlo Besta, 20133 Milan, Italy; paolo.ferroli@istituto-besta.it (P.F.); francesco.acerbi@unipi.it (F.A.); 6Department of Translational Research and New Technologies in Medicine and Surgery, University of Pisa, 56124 Pisa, Italy; 7Neurosurgery Unit, Pisa University Hospital, 56124 Pisa, Italy; 8Medical Genetics Unit, IRCCS Istituto Giannina Gaslini, 16147 Genoa, Italy; patriziademarco@gaslini.org

**Keywords:** moyamoya angiopathy, multiple sclerosis, demyelinating, neurovascular unit dysfunction

## Abstract

Moyamoya angiopathy (MMA) is a cerebrovascular disease determining chronic progressive steno-occlusion of the supraclinoid internal carotid arteries and their main branches. The pathogenesis of MMA remains largely unknown. Multiple sclerosis (MS) is a chronic, inflammatory, demyelinating disease of the central nervous system characterized by the progressive accumulation of focal demyelinating lesions, whose pathophysiology has been theorized but still incompletely understood. Beyond misdiagnoses due to mimicking features among the two disorders, MS coexisting with MMA have been previously, rarely, reported. Herein, we present two other cases of patients with MMA with a concomitant, previously missed, diagnosis of MS and discuss their overlapping features as a hint for a potentially shared pathophysiology. The finding of typical angiographic features enables MMA diagnosis, yet it does not allow us to rule out other potentially concomitant disorders affecting the CNS. The association may be easily missed if the clinical/neuroradiological picture is not carefully assessed. Cerebral spinal fluid analysis and spine neuroimaging should be suggested in all MMA patients with atypical MRI lesions.

## 1. Introduction

Moyamoya angiopathy (MMA) is a cerebrovascular disease determining chronic progressive steno-occlusion of the supraclinoid internal carotid arteries and their main branches. Its recognition is essentially based on specific angiographic findings: stenosis or occlusion at the terminal portion of the internal carotid artery (ICA) or proximal anterior cerebral artery (ACA) and/or middle cerebral artery (MCA) together with an abnormal vascular network (which in the most severe forms has been classically defined as a “puff of smoke”) at the base of the brain [1]. It is usually considered as a “non-inflammatory” condition; however, the pathogenesis of MMA remains largely unknown.

Multiple sclerosis (MS) is a chronic, inflammatory, demyelinating disease of the central nervous system characterized by the progressive accumulation of focal demyelinating lesions, whose pathophysiology has been theorized but still incompletely understood [2]. The exclusion of other diseases that can mimic MS is the cornerstone of current diagnostic criteria and the concept of “no better explanation” has been highly stressed for placing the diagnosis in recent years [3].

MMA and MS represent two separate conditions, but they also share some common clinical and radiological features. In addition, MS coexisting with MMA have been previously, although rarely, reported. Herein, we present two cases of concomitant MMA and MS and review the limited literature available on this topic.

## 2. Case Presentation

### 2.1. Case 1: A 45-Year-Old Woman Was Referred to Our Cerebrovascular Department for MMA

A medical history interview revealed anxiety and mood swings from puberty; endometriosis, leading to hysterectomy at the age of 43; and at the age of 17, she was evaluated for a transient episode of right-sided weakness, lasting approximately one week and spontaneously resolving. No medical investigations and diagnosis were made at that time.

No other neurological events were reported until the age of 45, when she experienced the first episode of blurred vision and diplopia. Brain magnetic resonance imaging (MRI) revealed multiple and bilateral small lesions, located in the left medulla and the right medullo-pontine passage, in the left subcortical white matter at the border between the middle and lower frontal gyrus, in the right superior temporal gyrus, in the left middle temporal gyrus, in both cerebellar hemispheres, and in the left periventricular region, without contrast enhancement. In addition, an MR angiogram revealed diminished flow voids in the middle and anterior cerebral arteries bilaterally, associated with prominent flow voids through the basal ganglia and thalamus due to a thick network of collateral vessels. Moreover, diminished cortical blood flow was inferred from fluid-attenuated inversion recovery (FLAIR) sequences showing a linear, sulcal high signal, which was compatible with an “ivy sign” [4]. The whole neuroradiological picture was thought to be consistent with a diagnosis of MMA.

CT angiography and perfusion, and a digital subtraction angiography (DSA), were performed and confirmed the diagnosis of MMA (Suzuki Grade IV) with evident collateral vessel networks also known as “puff of smoke sign”. Previous neurological symptoms and the presence of brain parenchymal lesions prompted to start specific treatment: antiplatelet therapy was initiated (with clopidogrel, due to allergy to acetylsalicylic acid). A two-step, bilateral surgical revascularization was also planned and performed, without complications.

Two years later, she complained of subacute onset of headache associated with diplopia, initially worsening, then partially improving over the following weeks. Neurological examination, performed after 3 weeks from the onset of the new symptoms, revealed horizontal diplopia from impaired left eye abduction. Gadolinium-enhanced brain MRI revealed a new T2-weighted hyperintense lesion in the left pontine tegmentum, without contrast enhancement (Figure 1). CT angiography, CT perfusion, and DSA were substantially unchanged from the previous ones; particularly, no signs of posterior circulation involvement were retrieved.

Due to the peculiar brainstem involvement and clinical course, to explore a possible inflammatory etiology, further examinations were then performed. Spine MRI and brainstem auditory, visual, and somatosensory evoked potentials were normal. Indeed, cerebral spinal fluid (CSF) analysis showed specific oligoclonal bands. Neuroimaging was then reviewed to identify features fitting with a diagnosis of demyelinating disease, as for the standard neuroradiological criteria for multiple sclerosis (MS) [3]. More than one supratentorial periventricular lesion >3 mm was considered compatible with demyelinating plaques due to their ovoid/elliptical morphology, being radially oriented, perpendicular to the ventricles without intervening with white matter (Figure 1C,D). Other iuxtacortical lesions were also considered compatible (Figure 1E). A lesion of the corpus callosum was also detected, although it was considered too small to be included in the neuroradiological assessment of spatial dissemination (Figure 1F). No spinal cord lesions were identified.

Therefore, the updated clinical picture, adding relapses to previous neurological history and reviewing brain parenchymal MRI findings, pointed to a coexisting diagnosis of MS, fulfilling McDonald’s criteria of dissemination in space and time.

Laboratory screening for neurological andsystemic autoimmune, and thrombophilic conditions revealed only a low titer for anti-double strand DNA antibodies (dsDNA Ab, 1:20).

The patient was treated with high-dose steroid pulses (intravenous methylprednisolone 1 gr OD for 5 days) with complete reversal of symptoms. Overall, a relapsing-remitting subtype was identified and subsequent treatment with teriflunomide was initiated.

Brain MRI with MR angiogram was repeated after 1 year. The multifocal cerebral lesions as well as the MMA neuroimaging features were stable, except for the area in the left dorsal pons that showed a decreased hyperintensity on T2-weighted images.

Whole exome sequencing was performed, and the following variants were identified (Table 1).

### 2.2. Case 2: A 43 Years-Old Woman Was Evaluated in Our Institute Because of the Neuroradiological Finding of Unilateral MMA

General medical history was unremarkable, whereas her neurological history had started at age 35. After experiencing post-infectious, self-remitting asthenia and limb paresthesias, she underwent a brain MRI that pointed out multiple, non-specific supratentorial lesions, with FLAIR and T2-w hyperintensity and without contrast enhancement. Neither angiogram nor further examination or intervention was made, except planning follow-up.

Two years later, she complained about two episodes of acute-onset, transient left hemi-anesthesia. Control brain MRI showed an increased number of supratentorial lesions (Figure 2). Angio-MRI revealed left MCA occlusion and distal ICA stenosis, which was considered expression of vascular dysplasia. Patent foramen ovalis (PFO) determining a severe right-to-left shunt was also detected and subsequently treated with endovascular closure. Without formulating a specific diagnosis but highlighting the presence of vascular risk factors in a patient with multifocal cerebral lesions, antiplatelet treatment with acetylsalicylic acid 100 mg daily was started.

Subsequent neuroimaging controls, performed over the next 6 years, showed a slow increase in the white matter lesion “burden”, with focal, slight contrast enhancement in some. Control angio-MRI performed at age 43 pointed out, besides the left steno-occlusion of the distal ICA and MCA, a deep network of collaterals; thus, fulfilling the criteria for unilateral MMA, she was then referred to our Institute. Recurrent episodes of transient left-sided sensory impairment were also reported.

The clinical and radiological evolution, despite the ongoing antiplatelet treatment and after PFO closure, urged a global revision of instrumental findings: the FLAIR/T2-w hyperintense lesions, multifocal and increasing over time, affecting especially the supratentorial regions but also the left cerebellar hemisphere and the bulbospinal junction, in some occasions associated with slight contrast enhancement, were finally considered as supporting an inflammatory, demyelinating etiopathophysiology. Further examinations were carried out. Laboratory screening for autoimmune diseases tested negative; among thrombophilic conditions, a heterozygosity for both MTHFR C677T and G20210A factor II was identified. CSF analysis disclosed specific oligoclonal bands. Spine MRI pointed out a single cervical (C7) lesion. A diagnosis of MS was then formulated, accounting for the evidence of dissemination in space and time, and treatment with dimethylfumarate was started.

Angiograms including vessel wall imaging failed to identify underlying or previous conditions pointing to MMA-mimics such as dissections or vasculitis. In the absence of any clinical symptoms properly referring to the unilateral MMA, perfusion MRI and acetazolamide-challenged SPECT were performed, documenting normal findings in both hemispheres. Revascularization surgery was then postponed, and long-term antiplatelet treatment confirmed.

The subsequent 8-year follow-up showed stable white matter burden both in brain and spine MRIs; MMA features have been confirmed as exclusively affecting the left carotid axis and are substantially stable over time. Dimethylfumarate was stopped after 5 years because of acquired lymphopenia, then shifted to Glatiramer acetate, then again shifted to Teriflunomide due to intolerance. The patient has not experienced any further clinical relapse but reported the onset of chronic fatigue. RNF213 sequencing was assessed and resulted wild-type (WT).

## 3. Discussion

Our reports describe the unusual concurrence of MS and MMA, where the diagnosis of MS was initially missed. While most reports linking MMA to MS have described misdiagnoses [6], the coexistence of the two disorders has been previously, albeit rarely, reported [6,7,8,9,10], raising the question of possible shared pathophysiological mechanisms (Figure 3; Table 2).

So far, no clear relationship between the two diseases has been elucidated.

Of note, MMA and MS share some clinical and neuroradiological features (Figure 3). First, both disorders typically affect young female patients. An association with other autoimmune diseases is also a common feature [2,11,12].

As a consequence of ischemia or hypoperfusion as well as for inflammatory demyelination, both MMA and MS can emerge with transient, often recurrent neurological symptoms, especially early in the disease course. In MMA, focal signs and symptoms are often abrupt (“ictal”), with a sudden or rapid increase in deficit, and when transient (if an actual stroke does not occur) they resolve within 24 h. In comparison, MS attacks progress subacutely over days before resolving over days-to-weeks, typically showing a more gradual and variable progression, with periods of worsening and remission [1,6].

MMA preferentially affects the anterior circulation, with symptoms usually reflecting the dysfunction of cerebral hemispheres, whereas demyelination may occur anywhere in the CNS, including the brainstem, cerebellum, optic nerve, and spinal cord. Finding specific neurological signs—as internuclear ophthalmoplegia, unilateral vision impairment, spinal sensory levels or hyperreflexia—suggest indeed to search outside the region of vulnerability in MMA [6].

“Stressors” such as intense physical activity, dehydration, and fever can trigger both ischemic events in MMA (being situations that may increase the hemodynamic demand) [13] and MS attacks due to Uthoff’s phenomenon [7,14].

Slow progression towards disability and cognitive impairment is also a common feature, often associated with psychological disorders [15]. MMA can manifest itself with other neurological symptoms, such as headache [16] and epilepsy [13], that have also been reported in MS patients, albeit considerably less frequently [6].

Multifocal FLAIR and T2-weighted subcortical hyperintense lesions on MRI are found in both, as expression of watershed ischemic damage or demyelination, respectively [17]. Demyelinating lesions are classically featured by ovoid/elliptical shape, that is considered expression of perivenular pathology [18]. Brainstem involvement is rather typical of CNS demyelination [19]; however, MMA, though unfrequently, may also affect posterior circulation [20]. In addition, demyelinating lesions can show DWI restriction in the acute phase, as well as acute ischemic foci can show post gadolinium enhancement, further complicating their differentiation. Furthermore, both conditions are featured by the involvement of the periventricular areas, being the classic location of demyelinating, white matter lesions in MS, and also the site where moyamoya collaterals (or—by using a more recent terminology—“periventricular anastomosis”, between the perforating or choroidal artery and medullary artery in the periventricular area) originate [21].

Misinterpreting the inflammatory lesions of MS as indicative of ischemic MMA lesions may lead to diagnostic and therapeutic delay and mistakes. The diagnosis of MS is currently made according to McDonald’s criteria, which requires, besides fulfilling criteria of disease dissemination in time and space, the presence of CSF-specific oligoclonal bands but also the exclusion of other potential aetiologies, as summarized by the concept of “no better explanation” [22,23,24]. Interestingly, rough application of this criterion can sometimes lead to an erroneous exclusion of demyelinating disease, when other possible causes are identified, even if these alternative causes have a much lower incidence than MS (which has been estimated on the order of 1:1000 in North American and European populations). In fact, in our cases, the first diagnosis was essentially driven by the angiogram findings. MMA typical angiograms are often considered peculiar, to the point that in their presence other potential causes of brain lesions tend to be overlooked.

However, it is also true that generally, if thorough examinations are not performed, the young adult, particularly if female, who presents with episodic neurological dysfunction and multifocal white matter lesions, is often assumed to be suffering from MS until proved otherwise [6].

In both of our cases, essentially masked by the early findings of the coexisting MMA, MS diagnosis was initially overlooked and important diagnostic examinations, such as CSF analysis or visual evoked potentials, were not performed. However, a post-hoc revision of the former imaging pointed out some clues that could have directed towards a suspicion of MS, such as the periventricular localization of some of the lesions or their elliptical/ovoid shape. Identifying either disease “onset” is almost impossible, though, in retrospect, clinical features could help in differentiating the expression of each disorder. In Case 1, the earliest symptoms were more suggestive of transient ischemic attacks than of MS poussées; otherwise, in Case 2, neurological symptoms could not be related to the ipsilateral angiopathy, thus more likely reflected expression of demyelination.

Ruling out MMA could be easier than ruling out MS, since the negativity of angiographic studies is sufficient to exclude the diagnosis; at least, at the time of evaluation. However, since MMA is a progressive condition, it cannot be excluded during patients’ follow up if angiographic studies are not repeated over time. Further data are warranted to assess if a subset of MS patients could develop MMA in the long term or vice versa, since after initial diagnostic differential processes, MS patients are not routinely re-assessed with angiograms over time, and MMA patients are explored with CSF analysis or spine MRI only in the presence of atypical features. In this view, a potential underestimation of this coexistence could be hypothesized as due to a “halo effect” from the first diagnosis, preventing the execution of thorough re-evaluations and instrumental examination over the course of time. Also, it is known that the long-term behavior of unilateral MMA (overall, including our experience, we identified unilateral MMA in three of the eight reported cases of associated disorders), including its potential evolution to bilateral MMA, is hard to predict and needs further studies [25], especially in the presence of coexisting inflammatory disorders, where the duration of one illness could theoretically play a role in promoting the evolution of the other one [26]. In this view, our Case 2 offers a notably extended follow-up of a strictly unilateral MMA for 15 years (calculating the first neuroimaging retrievable in the past history).

Misdiagnosis, especially under the “no better explanation” criterion, represents a notable risk in patients sharing common epidemiological features [6].

Interestingly, although MMA and MS may coexist by chance, a common pathogenic background could be hypothesized from clinical observations and pre-clinical evidence (Figure 3).

First, some rough similarities in epidemiology (female predominance, age range, existence of familial cases characterized by earlier onset) might suggest a common physiological milieu.

About genetic studies, which have been conducted in wide populations and familial cases to identify causative or susceptibility mutations, as far as our dissertation is concerned, a special mention should be reserved to the RNF213 gene. In fact, GWAS studies enabled RNF213 identification as a major candidate gene for MMA [27], especially the p.R4810K variant [28], although its involvement in disease pathophysiology has yet to be fully elucidated.

RNF213 is located on chromosome 17, encodes for a large cytosolic protein ubiquitously expressed, and contains a ring-finger E3 ubiquitin ligase domain and an AAA-ATPase domain. It has been found to be involved in angiogenesis, arterial wall remodeling, but also in chronic inflammation, nerve cell destruction, differentiation and formation of oligodendrocytes, and myelin depletion, thus resulting in a relevant position for both MMA and MS. Indeed, mutations of this gene have been found to be associated with an immunogenic role in patients with familial MMA and patients with familial MS [29,30]. In our experience, Case 2 carries RNF213 wild-type, whereas in Case 1 exome sequencing identified the RNF213 variant c.12092T>C, which is classified as a variant of unknown significance (VUS). Nevertheless, this variant, which to date has never been reported in MMA patients and in the databases of general population, affects the RING finger domain (aa 3397_4035), a hot spot region required for the correct ubiquitin ligase activity of the RNF213 protein. Interestingly, in the same case, two other very rare variants, classified as VUS according to standard guidelines, of genes involved in the cerebrovascular field have been identified: the c.2738C>T in the NOTCH3 gene and the c.1588C>T in the COL4A1 gene. Both NOTCH3 and COL4A1 are indeed associated with cerebral microvascular damage [31] and, although our patient currently does not present the most typical picture of any of the two syndromes, they can be considered when exploring the etiopathophysiology of small vessel disease and white matter lesions. Also, these findings could support the hypothesis of a “blended” phenotype in a carrier of multiple mutations or variants [32], as recently reported in another case of cerebrovasculopathy in a patient carrying concomitant NOTCH3 and RNF213 variants [33].

Inflammatory and immunological pathways are likely involved in both conditions, as suggested by altered levels of molecules such as cytokines, chemokines, and growth factors that have been observed in biological fluids of both patients with MMA [34,35,36], and patients with MS.

Also, the clinical observation of frequent association with other autoimmune disorders, such as Sjogren’s syndrome, Graves’ disease, rheumatoid arthritis, and systemic lupus erythematosus, which has been reported in both [14,37,38,39,40,41,42], may support the existence of immunopathological pathways, even without the involvement (or, at least, the identification) of specific antibodies.

Of note, in Case 1, another extremely low-frequency variant in control databases was identified in the JAK1 gene: c.1513G>A. This variant induces the change in a glycine with a serine at position 505 of the SH2 domain, involved in the protein–protein interaction and modulation of signaling cascade. Mutations of this gene, located on chromosome 1, have been associated with autoinflammatory and immune-mediated disorders [43]. In fact, JAK 1 belongs to the JAK family of intracellular, non-receptor tyrosine kinases that transduce cytokine-mediated signals via the JAK-signal transducers and activators of transcription (JAK-STAT) pathway. The JAK-STAT pathway is activated by many different pro-cytokine receptors that control hematopoiesis, immune cell differentiation and immunological response, embryogenesis, and inflammation through the signaling pathway. Besides neoplasms, many germline mutations or specific polymorphisms in JAK genes have been associated with autoimmune and inflammatory disorders, including MS [44,45].

MMA is generally considered as a non-inflammatory condition and differentiated from moyamoya-like angiographic appearance of well-defined vasculitic processes [46]. However, the role of neuroinflammatory pathways has been frequently hypothesized in its pathophysiology. In particular, it has been proposed that in specific genetic and acquired conditions, several factors such as infectious agents, immunological responses, flow-induced endothelial dysfunction such as shear stress, may be related to smooth muscle cell migration at internal carotid terminals and may trigger the development of arteriopathy (double hit hypothesis) [47,48].

The influence of chronic inflammation on the pathophysiology of MMA is still under debate. The elevated levels of inflammatory molecules such as cytokines, chemokines, and growth factors and deficient Treg suppressive functions that have been found in patients with MMA [35] may be considered causative, as well as result of the condition itself. In fact, anti-inflammatory and pro-inflammatory cytokines have been suggested to influence the activation of the RNF213 gene, which has been strongly related to MMA pathogenesis. Particularly, interleukin (IL)-4, IL-10, IL-13, interferon (IFN)-alfa, and transforming growth factor (TGF)-beta induce M2 macrophages, which are responsible for promoting angiogenesis. Furthermore, IFN-beta, IFN-gamma, tumor necrosis factor (TNF)-alfa, IL-6, and IL-1 may activate synergistically the transcription of RNF213.

These pro-inflammatory cytokines seem to be crucial also in the pathogenesis of MS and have been found to influence the recruitment of inflammatory cells into the CNS, responsible for the neuronal demyelination and the induction of inflammation within the CNS parenchyma. A higher RNF213 expression and pathogenic variants affecting transcription and activation of inflammatory mediators have also been described in familial cases of multiple sclerosis [30,49].

Due to these similarities and shared molecular basis, some authors hypothesized an interaction between MMA and MS, with MMA potentially triggering or accelerating demyelination processes, or MS putatively inducing a vasculitic phenomenon, finally leading to development of MMA [8] (Figure 3).

Another linking point could be represented by the role of neurovascular unit dysfunction in both disorders.

Neurovascular unit dysfunction is obviously theorized in MMA pathophysiology, because the disease essentially determines progressive impairment to cerebral blood perfusion, with a particular susceptibility to hypoperfusion due to ineffective collateral compensation.

MS hallmark is classically represented by demyelination; however, several studies described also the existence of significant vascular pathology in MS, ranging from reduced blood flow in both white and gray matter, chronic cerebral hypoperfusion, and hypoxia to reduced cerebrovascular reactivity [50,51]. In fact, MS has vascular risk factors: the major environmental and lifestyle risk factors MS onset include vitamin D deficiency [52,53], cigarette smoking [54], and youth obesity [55,56].

In MS, many histopathological reports have highlighted the presence of damaged blood vessels due to loss of endothelial cell tight junctions and pericytes, detachment and consequent activation of astrocytes, and vessel disruption that is operated by oligodendrocytes progenitor cells (OPCs). Indeed, OPCs participate both in myelination [57] and neurovascular unit functions, in the latter case by regulating angiogenesis, through VEGF-A secretion, and BBB integrity, especially under hypoxic conditions. In addition, evidence of endothelial cell proliferation and increased vessel density in MS studies suggest phenomena of extensive angiogenesis [58]; however, there is evidence for the formation of incomplete neurovascular units, ultimately impairing neurovascular signaling and metabolic demand, further contributing to oligodendrocyte death [59,60]. Furthermore, neurovascular damage may lead to the leakage of blood-derived products, including fibrinogen and leukocytes, which may cause microglial activation, proliferation, and migration of perivascular fibroblasts from the Virchow-Robin space, inflammation, oligodendrocyte death, demyelination, defective remyelination and neurodegeneration [50]. Gene expression analysis of intra-lesional microvessels from patients with MS showed an increased expression of inflammatory factors including MMP (matrix metalloproteinase)2, MMP14, ADAM17 (disintegrin and metalloproteinase domain-containing protein 17), VEGF-A, and VEGFR1 (vascular endothelial growth factor receptor 1) [61], suggesting that inflammation could trigger cerebral hypoperfusion and hypoxia [62,63,64]. Overall, it is unclear whether dysfunction of key cells within the neurovascular unit is primary or secondary in MS pathophysiology.

Interestingly, another demyelinating immune-mediated disease specifically targeting the CNS has also been reported in association to MMA: neuromyelitis optica (NMO) (Table 3).

NMO typically involves optic nerves and the spinal cord, the latter with lesions extending over several segments; the involvement of area postrema, brainstem, and diencephalic structures has also been observed. In affected patients, specific autoantibodies, NMO IgG, bind Aquaporin-4 (AQP4), the main water homeostasis channel in the CNS, and are thought to induce the infiltration of brain tissues by inflammatory cells through the activation of classic complement pathway, damaging astrocytes, oligodendrocytes, and neurons [65]. In addition, NMO cases are often featured by the finding of anti–SS-A, anti-SS-B, and anti–dsDNA antibodies, which are also found in patients with Sjogren’s syndrome (SS) and have been hypothesized to participate in inducing the vascular damage potentially ending in MMA [26,66,67]. In our Case 1, only a low titer of anti-dsDNA antibodies was detected.

Our reports raise two important diagnostic and therapeutic concepts:-On one hand, it is indeed imperative to diagnose and promptly treat MMA at the early stages with antiplatelets and revascularization surgery, before irreversible stroke occurs.On the other hand, the definite diagnosis of MMA should not automatically exclude MS as a possible concomitant disease, especially considering the higher incidence of MS [6] and the role that autoimmune mechanisms potentially play in both these conditions. Thus, it may be reasonable to recommend CSF analysis and spine MRI in the diagnostic work-up of MMA, especially in the presence of atypical features, e.g., when MRI lesions are not exclusively located in the typical borderzone territories (‘watershed lesions’) and/or when their morphology may suggest a different etiology.

**Table 3 ijms-26-05030-t003:** Comparison of cases from literature reporting the association of MMA and NMO.

Case	Country/ Ethnicity	Age/Sex	Clinical Presentation	T2-w MRI White Matter Hyperintensities Location	Gadolinium Enhancement	Spine MRI	CSF OB	VEP	Other Tests	Angiogram Findings	Diagnosis	Any Clinical Stroke?	Treatment		FU
													For MMA	For NMO	
[66]	Japan/ n.s.	52/F	Blindness, sensory-motor impairment lower limbs, urinary retention	Not specified (Hemorrhage in right thalamus) Longitudinal thoracic spinal cord	Brain: − Spine: +	+	−	+	ANA (+ 1:80) Anti SS-A/B Ab (+) AQP4 Ab (+) ANCA (−) Anticardiolipin (−)	Bilateral ACA and MCA occlusion, bilateral MM network	MMA + NMO + Systemic sclerosis	Hemorrhage	n.s. (hemorrhagic)	High dose i.v. corticosteroid pulse	4 month
[67]	Hong Kong/East Asian	62/F	Gait impairment to tetraparesis and sensory impairment	Brain: − Longitudinal cervico-thoracic (C2-T3) lesions	Spine: n.s.	+	n.s.	+	Anti SS-A Ab (+) AQP4 Ab (+) ANA (−)	Bilateral ICAD and MCA occlusion, bilateral MM network	MMA + NMO	n.s.	n.s.	High dose i.v. corticosteroid pulse, followed by oral tapering and azathioprine, followed by cyclophosphamide	8 month
[26]	China/ n.s.	43/F	Left sensory impairment	Brain: centrum semiovale + DWI restriction in right thalamus, hemosiderin deposition in left basal ganglia Thoracic spinal cord	Brain: n.s. Spine: n.s.	+	n.s.	+	AQP4 Ab (+)	Bilateral ACA and MCA, right PCA and right ICAD stenosis, bilateral MM network	MMA + NMO	Hemorrhagic and ischemic	(hemorrhagic) Antiplatelet	I.v. and oral corticosteroids, azathioprine	6 month

Abbreviations: ICA, Internal Carotid Artery (ICAD, ICA-Distal). MCA, Middle Cerebral Artery. ACA, Anterior Cerebral Artery. PCA, Posterior Cerebral Artery. FU, Follow-up. CSF, Cerebrospinal Fluid. OB, Oligoclonal Bands. VEP, Visual Evoked Potential. MM(A), Moyamoya (angiopathy). NMO, Neuromyelitis optica. Ab, Antibodies, anti-: AQP4, Aquaporin-4. ANA, Antinuclear Antibodies. ANCA, Antineutrophil Cytoplasmic Antibodies. n.s., not specified; as for the reported diagnostic investigations, results are simplified as positive (+) if depicting diagnostic or at least supporting alterations, otherwise negative (−).

## 4. Materials and Methods

A literature review was conducted by searching the relevant keywords on Pubmed: “moyamoya” AND “multiple sclerosis” OR “demyelinating” OR “demyelination” OR “neuromyelitis optica” OR “inflammatory” OR “inflammation”.

## 5. Conclusions

In conclusion, MMA and MS can coexist. Moreover, CNS demyelination cannot be ruled out by the diagnosis of concomitant MMA: MS may be easily missed if imaging studies and neurological history and examination are not assessed and reviewed carefully. Our cases and the results of our literature review highlight the need of considering other possible diagnoses in cases of atypical MMA signs and symptoms and suggest a potential underestimation of the number of patients affected by these coexisting conditions. Given the possible common inflammatory pathways, MS may be considered among the immunological diseases associated with MMA. Further studies and a long-term follow-up of the reported cases are necessary to clarify the pathophysiological relationship between the two conditions.

## Figures and Tables

**Figure 1 ijms-26-05030-f001:**
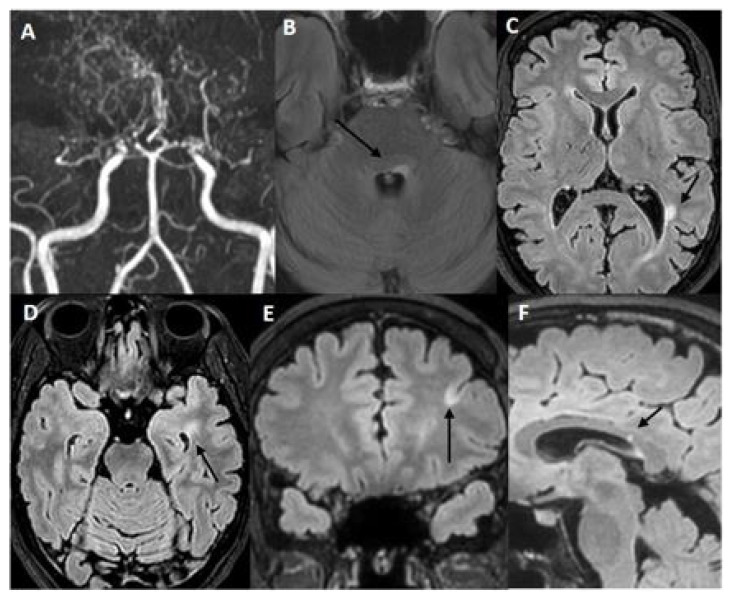
Case 1—Brain neuroimaging. (**A**) MR angiography: bilateral stenosis of the supraclinoid internal carotid arteries with multiple collateral moyamoya vessels. (**B**–**F**) Multiplanar FLAIR sequences depict multifocal demyelinating lesions (arrows): left pontine lesion (**B**), periventricular lesions (**C**,**D**), frontal juxtacortical lesion (**E**), intracallosal lesion (**F**).

**Figure 2 ijms-26-05030-f002:**
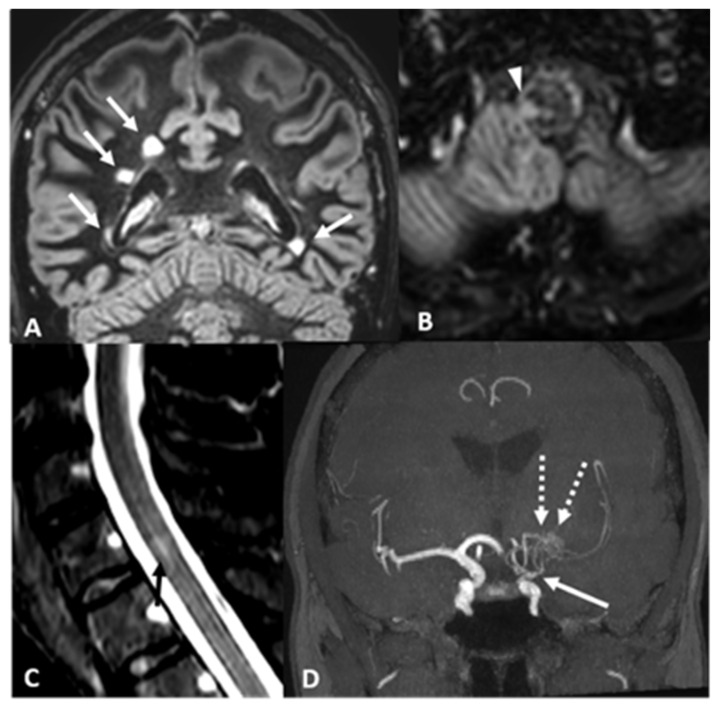
Case 2—Brain and spine imaging. (**A**) Coronal and (**B**) axial DIR sequences show multiple bilateral MS-typical periventricular lesions (arrows) perpendicular to the lateral ventricle walls, as well as a small lesion in the right part of the bulbo-medullary junction (arrowhead in (**B**)). (**C**) Sagittal MR STIR sequence of the cervical spine reveals a faint hyperintensity in the cervical cord (arrow). (**D**) TOF AngioMR, MIP reconstruction, confirms unilateral left MMA with ICA stenosis (arrow) and compensatory hypertrophy and hyperplasia of the lenticulostriate arteries (dotted arrows).

**Figure 3 ijms-26-05030-f003:**
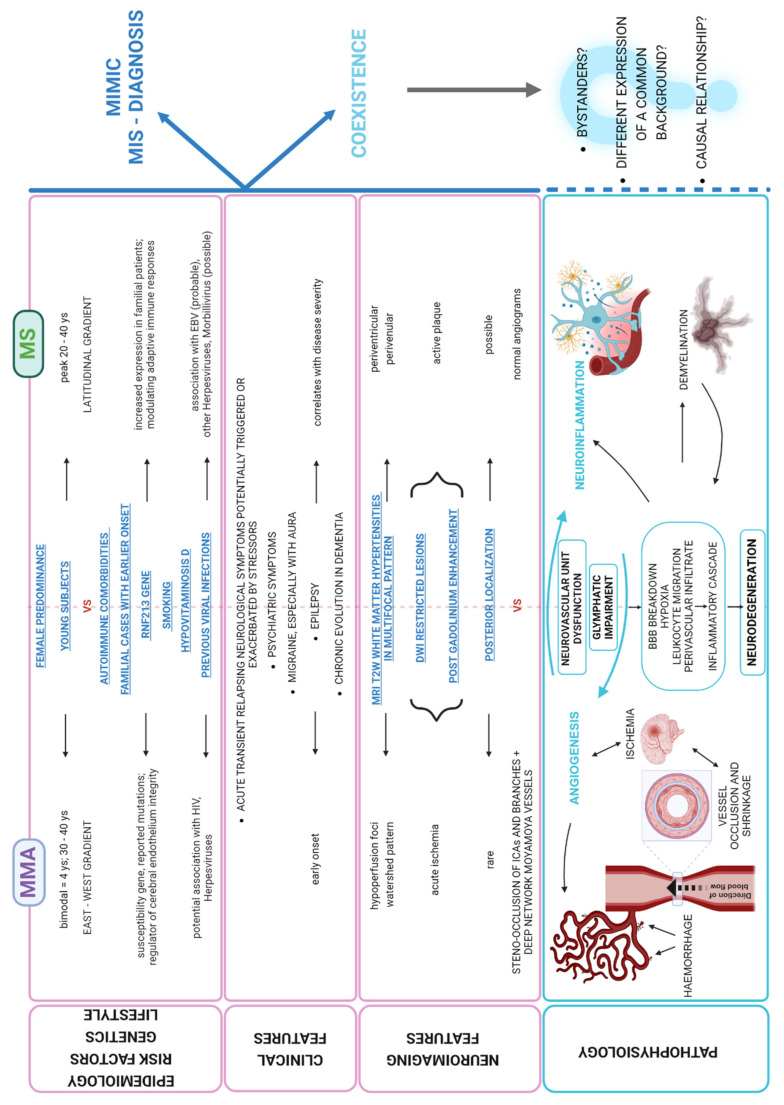
The complex relationship between MMA and MS: similarities in epidemiology, clinical and neuroimaging expression may act as confounders as well as suggest shared pathophysiological pathways or at least a common background.

**Table 1 ijms-26-05030-t001:** Results from whole exome sequencing performed on Case 1’s sample.

Gene (Transcript)	Nucleotide Change	Aminoacid Change	ACMG Class *	Disease (OMIM)
*RNF213* (NM_001256071.3)	c.12092T>C	p.Ile4031Thr	III (VUS)	Moyamoya (*607151)
*NOTCH3* (NM_000435.3)	c.2738C>T	p.Pro913Leu	III (VUS)	Cerebral arteriopathy (*600276)
*COL4A1* (NM_0018450.6)	c.1588C>T	p.Pro530Ser	III (VUS)	Microangiopathy and leukoencephalopathy, pontine, autosomal dominant (*618564)
*JAK1* (NM_002227.4)	c.1513G>A	p.Gly505Ser	III (VUS)	Autoinflammation, Immune dysregulation (*618999)

* ACMG, American College of Medical Genetics [5]. VUS: variant of uncertain significance.

**Table 2 ijms-26-05030-t002:** Comparison of cases from literature reporting the association of MMA and MS (one case [7] may depict an association of the two disorders but does not rule out their misdiagnosis).

Case	Country/Ethnicity	Age/ Sex	Clinical Presentation	T2-w MRI White Matter Hyperintensities Location	Gadolinium Enhancement	Spine MRI	CSF OB	VEP	Other Tests	Angiogram Findings	Diagnosis	Any Clinical Stroke?	Treatment		FU
													For MMA	For MS	
[6] (case 3)	US/ n.s.	44/F	Intermittent left hemiparesis	Multiple subcortical lesions (n.s.)	n.s.	−	+	−	n.s.	Left ICA occlusion, severe right ICAD stenosis, bilateral MM network	MMA + MS	Ischemic stroke	Bilateral bypass surgery, Aspirin	b-interferon, glatiramer acetate, high dose i.v. corticosteroids	6 month
[6] (case 9)	US/ n.s.	31/F	Limb numbness and weakness, memory impairment	Multiple subcortical lesions (n.s.)	n.s.	n.s.	n.s.	n.s.	n.s.	Right MCA occlusion, R ACA stenosis	MMA + MS	n.s.	Right bypass surgery	n.s.	18 month
[7] (case 2)	US/ n.s.	44/M	Right hemiparesis; impaired vision	Periventricular	Brain: −	n.p.	−	+	ANA (+)—ENA (−)	Left ACA, MCA and PCA stenosis	MMA mimicking/ associated with MS (discussion not conclusive)	TIA	Aspirin + Clopidogrel	b-interferon (stopped after MMA evidence)	n.s.
[8]	China/ n.s.	42/M	Right hemiparesis, speech impairment	Cervical spinal cord; Frontal, parietal, temporal lobe	Brain: + Spine: +	+	+	+	Anti-b2-glycoprotein I IgA (++)	Bilateral ICAD and right MCA occlusion; Left ACA, MCA, PCA stenosis	MMA + MS	Ischemic stroke (during steroid administration)	Aspirin + Clopidogrel; bilateral bypass surgery	High dose i.v. methylprednisolone	3 month
[9]	US/ Caucasian	57/F	Episodic gait dysfunction; feet paresthesias; myelopathic signs	Cervical and thoracic spinal cord; bilateral frontal lobes; right pons; right cerebellum	Brain: spotty + Spine: −	+	+	NA	ANA (+ 1:80) Ab anti AQP4 and anti MOG (−) Ab anti cardiolipin (−)	Bilateral MCA and ACA occlusion, MM network	MMA + MS	No	n.s.	n.s.	n.s.
[10]	Saudi Arabia/ n.s.	16/F	Left hemiparesis	Periventricular lesion	n.s.	n.s.	+	n.s.	n.s.	Bilateral steno-occlusion ICAs-MCAs, R-ACA and R-PCA stenosis, lentriculostriate network	MMA + MS	Ischemic stroke (2)	Aspirin; bilateral by-pass surgery	Interferon beta and steroids	n.s.
Our Case 1	Italy/ Caucasian	45/F	Blurred vision and diplopia	Bilateral brainstem, left frontal and temporal lobe, right temporal lobe, cerebellar hemispheres, left periventricular region	Brain:	−	+	−	AntiAQP4 (−) Anti-ds DNA (+ 1:20) ANA-ENA-ANCA-Anti cardiolipin and beta2GP (−)	Bilateral MCAs and ACAs narrow, thick network of collateral vessels	MMA + MS	TIA	By-pass surgery, Aspirin	Corticosteroids Teriflunomide	4 years
Our Case 2	Italy/ Caucasian	43/F	Left hemi-anesthesia (transient, recurring)	Bilateral supratentorial periventricular with corpus callosum involvement, left cerebellar hemisphere; bulbospinal tract; cervical spine	Brain: − Spine: −	+	+	−	AntiAQP4 (−) Anti-ds DNA-ANA-ENA-ANCA-Anti cardiolipin and beta2GP(−) G20210A Factor II and MTHFRC1677T (heterozigosis) Total body PET (−)	Left MCA and distal ICA steno-occlusion, MM network	MMA + MS	no	Aspirin	Dimethylfumarate Glatiramer acetate Teriflunomide	8 years

Abbreviations: ICA, Internal Carotid Artery (ICAD, ICA-Distal). MCA, Middle Cerebral Artery. ACA, Anterior Cerebral Artery. PCA, Posterior Cerebral Artery. FU, Follow-up. CSF, Cerebrospinal Fluid. OB, Oligoclonal Bands. VEP, Visual Evoked Potential. MM(A), Moyamoya (angiopathy). Ab, Antibodies, anti-: AQP4, Aquaporin-4; MOG, Myelin Oligodendrocyte Glycoprotein; ds DNA, Double-Stranded Deoxyribonucleic Acid. ANA, Antinuclear Antibodies. ENA, Extractable Nuclear Antigens. ANCA, Antineutrophil Cytoplasmic Antibodies. PET, Positron Emission Tomography. n.s., not specified; n.p., not performed; as for the reported diagnostic investigations, results are simplified as positive (+) if depicting diagnostic or at least supporting alterations, otherwise negative (−).

## Data Availability

The original contributions presented in this study are included in the article. Further inquiries can be directed to the corresponding authors.

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
