# Peer review of "Blurred by a “Puff of Smoke”—A Case-Based Review on the Challenging Recognition of Coexisting CNS Demyelinating Disease and Moyamoya Angiopathy"

_ijms, 2025, doi:10.3390/ijms26115030_

Round 1
Reviewer 1 Report
Comments and Suggestions for Authors
Dear Authors,
I would like to congratulate you for this paper, which is, to my opinion, of high clinical value especially considering the possible outcome of the patients without correct diagnosis and therapy. Although it has already been described, the incidence of those two conditions is rare, but extremely important to keep in mind.
From my side I do not have any comments or corrections to point out.
Author Response
We are sincerely grateful for Your kind appreciation and are happy that You consider our work interesting, adequate and above all useful for clinicians.
Thanks, again!
Reviewer 2 Report
Comments and Suggestions for Authors
The authors present two cases of co-occurrence of moyamoya angiopathy and multiple sclerosis. The article is interesting and reflects the difficulties of differential diagnosis of these disorders. The only suggestion concerns a possible more precise description of the rate of increase of deficit in the case of vascular symptoms and those resulting from demyelination.
Author Response
"The authors present two cases of co-occurrence of moyamoya angiopathy and multiple sclerosis. The article is interesting and reflects the difficulties of differential diagnosis of these disorders".
We sincerely thank the Reviewer for his kind report and positive comments.
"The only suggestion concerns a possible more precise description of the rate of increase of deficit in the case of vascular symptoms and those resulting from demyelination"
We thank the Reviewer for this suggestion. We have tried to include some additional information to help differentiate the clinical course of vascular versus demyelinating symptoms:
"As a consequence of ischemia or hypoperfusion as well as for inflammatory demyelination, both MMA and MS can present with transient, often recurrent neurological symptoms, especially early in the disease course. In MMA, focal signs and symptoms are often abrupt ("ictal") with a sudden or rapid increase in deficit, and when transient (if an actual stroke does not occur) they resolve within 24 hours. In comparison, MS attacks progress subacutely over days before resolving over days-to-weeks, typically showing a more gradual and variable progression, with periods of worsening and remission [1,5]".
We hope that You will find our work sufficiently improved after revision.
Reviewer 3 Report
Comments and Suggestions for Authors
Review comments are attached
The manuscript describes the concurrent evaluation of two pathological conditions; Moyamoya Angiopathy (MMA) and Multiple Sclerosis (MS). To make the point that due care should be given when one or both of the conditions are diagnosed correctly, the clinical findings in each case are presented in a way that they formulate a global profile discriminating between the two conditions, a fact often disregarded. To that end, misdiagnosis of one or the other disorder could easily occur, thereby precluding decisions linked to therapeutic administration of appropriate pharmaceuticals. The thorough examination of the two specific cases and the underlying causes of the pathology(ies) are provided to the extent that they describe the genetics, triggering events as well as potential links that could lead to common pathways or parallel pathways setting on the disease.
The work was run competently, yet there are several parts in the manuscript that require revisions prior to any consideration. In that respect, a select list of such ill-understood cases of description is provided fir further consideration.
1. In the introduction, the statement “Two years later, she complained the subacute onset of headache associated with diplopia, initially worsening than partially improving over the following weeks.” should rather be rewritten to read “Two years later, she complained of subacute onset of headache associated with diplopia, initially worsening then partially improving over the following weeks.”.
2. In line 113, the correct name of the drug is methylprednisolone.
3. In lines 158-159, the statement “Further examinations were then acquired.” should be written to tread “Further examinations were carried out.”.
4. In lines 216-217, the statement “Finding specific neurological signs - as internuclear ophthalmoplegia, unilateral vision impairment, spinal sensory levels or hyperreflexia …” should be rewritten to read “Finding specific neurological signs - as internuclear ophthalmoplegia, unilateral vision impairment, spinal sensory levels or hyperreflexia …”.
5. In lines 219-227, the entire two paragraphs are not correctly dwelled on. To that end, ““Stressors” such as intensive physical activity, dehydration, fever can trigger both ischemic events in MMA (being situations that may increase the hemodynamic demand) [12] and MS attacks due to Uthoff’s phenomenon [6,13]; the identification of such events in medical history can be supportive in both these conditions but, again, not discriminating among them.
A slow progression towards disability and cognitive impairment is also a common feature, often associated with psychological disorders [14]. MMA can present with other neurological symptoms, such as headache [15] and epilepsy [12], that have been reported also in MS patients but much less commonly [5].” should be rewritten to read “Stressors”, such as intense physical activity, dehydration, fever can trigger both ischemic events in MMA (being situations that may increase the hemodynamic demand) [12] and MS attacks due to Uthoff’s phenomenon [6,13]; the identification of such events in medical history can be supportive of both of these conditions but, again, not discriminating between them.
Slow progression toward disability and cognitive impairment is also a common feature, often associated with psychological disorders [14]. MMA can be present with other neurological symptoms, such as headache [15] and epilepsy [12], that have also been reported in MS patients, albeit considerably less commonly [5].”.
1. In lines 231-233, the statement “Brainstem involvement is rather typical of CNS demyelination [18], however also MMA, though unfrequently, may affect posterior circulation, enhancing misdiagnoses [19].” should be rewritten to read “Brainstem involvement is rather typical of CNS demyelination [18], however MMA, though infrequently, may also affect posterior circulation, enhancing misdiagnoses [19].”.
2. In lines 246-248, the statement “Interestingly, a rough application of this criterion can sometimes lead
to erroneously exclude the diagnosis of demyelinating disease when other possible causes are identified, even …” should be rewritten to read “Interestingly, rough application of this criterion can sometimes lead to an erroneous exclusion of demyelinating disease, when other possible causes are identified, even …”.
3. In lines 260-262, the statement “In both our cases, essentially shaded by the early findings of the coexisting MMA, the diagnosis of MS was initially overlooked and important diagnostic examinations, such as CSF analysis or visual evoked potentials, were not performed.” could be rewritten to read “In both of our cases, essentially masked by the early findings of the coexisting MMA, MS diagnosis was initially overlooked and important diagnostic examinations, such as CSF analysis or visual evoked potentials, were not performed.”.
4. In lines 267-269, the statement “otherwise, the neurological symptoms reported in case 2 could not be related to the contralateral angiopathy, thus were much more likely expression of demyelination.” could be corrected to read “otherwise, the neurological symptoms reported in case 2 could not be related to the contralateral angiopathy, thus they more likely reflected expression of demyelination.”.
5. In lines 388-390, the statement “Indeed, OPCs participate both to myelination [56] and to
neurovascular unit functions, the latter by regulating angiogenesis, through VEGF-A secretion, and BBB integrity, especially in hypoxic conditions. In addition, evidence of …” could be better expressed as “Indeed, OPCs participate both in myelination [56] and neurovascular unit functions, in the latter case by regulating angiogenesis, through VEGF-A secretion, and BBB integrity, especially under hypoxic conditions. In addition, evidence of …”.
Based on the aforementioned remarks, the manuscript should be revised and re-evaluated.

Review comments are attached
Author Response
- See attachment for detailed Reviewer's report -
We sincerely thank the Reviewer for the positive comments, the careful evaluation of our paper and especially for including precise corrections of our text, that we included as suggested throughout the revised version of the manuscript.
We hope that after fixing the text as suggested, You will consider our work sufficiently interesting and useful for clinicians, and worthy for Your re-evaluation.
Reviewer 4 Report
Comments and Suggestions for Authors
An interesting description on the association of multiple sclerosis with moya-moya angiopathy in 2 patients. Despite this association could be due to chance, the authors developed an interesting hypothesis. However, discussion is too long, and could be considerably shortenned without loosing information. The abstract could be shortenned as well (specifically, at least two paragraphs are nearly identical to other paragraphs in the introduction)
Author Response
"An interesting description on the association of multiple sclerosis with moya-moya angiopathy in 2 patients. Despite this association could be due to chance, the authors developed an interesting hypothesis."
We sincerely thank the Reviewer for appreciating the value of our work and discussion.
"However, discussion is too long, and could be considerably shortenned without loosing information"
We thank the Reviewer for the advice. We have now tried to shorten the discussion, especially by removing redundant expressions and descriptions. However, some paragraphs have been only slightly changed to take into consideration also the comments submitted by the other Reviewers. We hope that after the revision the Reviewer will find the discussion sufficiently improved.
"The abstract could be shortenned as well (specifically, at least two paragraphs are nearly identical to other paragraphs in the introduction)"
We thank the Reviewer for pointing this out. We have tried to shorten the Abstract as well, by removing similar paragraphs and avoiding repeated messages. We hope that the Reviewer will appreciate the changes we have made to the manuscript.
Round 2
Reviewer 3 Report
Comments and Suggestions for Authors
Review comments are attached
The revised manuscript is an improved version of the originally submitted one. However,
several points are still needed to be attended to. Remarks on specific statements are provided
below:
- In lines 229-231, the statement “As a consequence of ischemia or hypoperfusion as well
as for inflammatory demyelination, both MMA and MS can present with transient, often
recurrent neurological symptoms, especially early in the disease course.” should be
rewritten to read “As a consequence of ischemia or hypoperfusion as well as for
inflammatory demyelination, both MMA and MS can emerge with transient, often
recurrent neurological symptoms, especially early in the disease course.”.
- In lines 313-319, the statement “In Case 1, although it is not possible to specify the
onset of either, but in retrospect the earliest symptoms were more suggestive of
transient ischemic attacks than of MS poussées; otherwise, the neurological symptoms
reported in Case 2 could not be related to the contralateral angiopathy, thus more likely
reflected expression of demyelination” should be rewritten. Structurally, it is too long
and chaotic. It’s linguistic rendition should be improved.
- In lines 246-248, the statement “MMA can present with other neurological symptoms,
such as headache [15] and epilepsy [12], that have also been reported in MS patients,
albeit considerably less commonly [5].” should be rewritten to read “MMA can
manifest itself with other neurological symptoms, such as headache [15] and epilepsy
[12], that have also been reported in MS patients, albeit considerably less frequently [5].”.
Based on the above remarks, the manuscript should be revised. After that is accepted.

Review comments are attached
Author Response
- See Reviewer's report attached -
We sincerely thank, again, the Reviewer for the precious and really accurate revision of our manuscript, including the very appreciated suggested correction of the text, that we have included in the revised version of the paper we are uploading.
We are truly grateful of the positive comments and the opportunity of further improving the quality of our work.